# The Efficacy of Lactulose for the Treatment of Hyperammonemic Encephalopathy Due to Severe Heart Failure

**DOI:** 10.3390/diagnostics10020070

**Published:** 2020-01-27

**Authors:** Koichi Narita, Eisuke Amiya, Nobutaka Kakuda, Hidehiro Kaneko, Masaru Hatano, Issei Komuro

**Affiliations:** 1Department of Cardiovascular Medicine, Graduate School of Medicine, The University of Tokyo, Hongo 7-3-1, Bunkyo-ku, Tokyo 113-8655, Japan; knarita27@gmail.com (K.N.); kakudan-int@h.u-tokyo.ac.jp (N.K.); kanekohidehiro@gmail.com (H.K.); hatanoma@pg8.so-net.ne.jp (M.H.); komuro_tky2000@yahoo.co.jp (I.K.); 2Department of Therapeutic Strategy for Heart Failure, The University of Tokyo, Hongo 7-3-1, Bunkyo-ku, Tokyo 113-8655, Japan; 3The Department of Advanced Cardiology, The University of Tokyo, Hongo 7-3-1, Bunkyo-ku, Tokyo 113-8655, Japan

**Keywords:** heart failure, hyperammonemic encephalopathy, congestion, lactulose

## Abstract

Hyperammonemic encephalopathy secondary to heart failure is rare and there had been little reports about effective treatment. Organ hypoperfusion or congestion by heart failure may lead to various organ dysfunctions, and liver and intestinal circulatory impairment might cause ammonia metabolic failure. Here, we report on the case of a patient with hyperammonemic encephalopathy that was secondary to heart failure, which was effectively treated by lactulose.

## 1. Introduction

Heart failure is associated with various organ dysfunctions such as hepatic or renal dysfunctions due to the impairment of organ perfusion [1,2]. Hepatic dysfunction in congestive heart failure is occasionally encountered, revealing some aspects of the condition [3]. The complication is generally characterized by elevated bilirubin, alkaline phosphatase, and transaminase, with a prolonged prothrombin time and, sometimes, hyperammonemia. However, there are few cases where complications due to hyperammonemic encephalopathy occurs. Ammonia is mainly produced in the gut as a byproduct of protein digestion [4], bacterial metabolism [5], and liver metabolism. Therefore, hypoperfusion or congestion due to heart failure may lead to increased abdominal and liver damage, which might lead to the failure of ammonia metabolic balance [6]. However, there are scanty reports on cases of hyperammonemic encephalopathy associated with heart failure.

Here, we report on a case of heart failure with hyperammonemic encephalopathy, which was treated successfully by the addition of lactuloses.

## 2. Case Presentation

An 81-year-old male, who had suffered from heart failure with moderate to severe mitral regurgitation (MR), was hospitalized at our hospital. One year ago, he had been hospitalized for worsening heart failure, but did not wish to undergo valve surgery, hence, was followed up with medical therapy including beta-blockers and diuretics at the Outpatient Department. Right heart catheterization at that time revealed slight venous congestion (mean right atrial pressure; 12 mmHg) without low output and the increase of pulmonary capillary wedge pressure. The complication of membrane nephropathy was diagnosed 20 years ago, though his renal function was moderately decreased leading to a slight increase in blood urea nitrogen (BUN) and creatinine (Cre). Despite the medication for his heart failure, there had been slight increases in the concentration of total bilirubin (1.5–2.3 mg/dL), BUN, and Cre (1.6–2.2 mg/dL), partly due to medically intractable hepatic and renal congestion. In addition, his level of consciousness and physical activities had gradually decreased, which led to his hospitalization. His social history included some alcohol intake, which had significantly decreased in the last few years. On admission, the physical findings revealed flapping tremors, which is an involuntary low amplitude movement induced by actions such as hyperextension of the fingers keeping the wrist joint bent. His laboratory tests were as follows: white blood cell counts 3000/μL, hemoglobin 9.3 g/L, platelet counts 82,000/μL, BUN 56.3 mmol/L, Cre 1.67 μmol/L, bilirubin 2.0 μmol/L, albumin 3.2 g/L, aspartate aminotransferase (AST) 36 U/L, alanine aminotransferase (ALT) 17 U/L, alkaline phosphatase 760 U/L, C reactive protein 0.21 mg/dL, and prothrombin time of 28.4 s. These data showed that there had been a little change in his indices within the previous weeks. However, we checked the blood ammonia level for the first time and it was as high as 221 μg/dL. Electrocardiogram revealed atrial fibrillation, but there was no evidence of myocardial ischemia or infarction (Figure 1). Chest x-ray showed remarkable cardiac enlargement (cardio-thoracic ratio: 88%) (Figure 2). Echocardiogram demonstrated preserved ejection fraction and severe MR in addition to moderate tricuspid regurgitation (Figure 3), which were also observed four months before this time. The examination of his right heart function revealed that tricuspid annular plane systolic excursion of 18 mm, and right ventricle (RV) S’ of 12.4 cm/s, RV E’ of 9.0 cm/s, and septal E/E’ of 21.7 in tissue doppler imaging. As a differential diagnosis of consciousness disorder, there were no abnormal findings such as cerebral edema in the head computed tomography (CT) (Figure 4). An abdominal CT showed chronic changes in long-term liver damage due to liver congestion derived from right heart failure, but there were no signs of liver cirrhosis (Figure 5). Hepatic disorders including autoimmune, and alcoholism were negative (anti-nuclear antibody; weakly positive at a serum dilution of 1:40, anti-mitochondria antibody; negative). We had confirmed negative for hepatitis B virus (HBV) surface antigen, antibody, and hepatitis C virus (HCV) antibody leading to no suspect of the possibility of viral hepatic disorders. His thyroid function was as follows: Thyroid stimulating hormone 4.64 μIU/mL, free triiodothyronine 1.4 pg/mL, and free thyroxin 1.6 ng/dL. He was diagnosed with hyperammonemic encephalopathy. We first started the administration of lactulose (60 mL (39 g) per day) in order to suppress ammonia production in the intestinal tract, which resulted in a significant decrease of NH3 ((221 μg/dL(baseline)→158 μg/dL (10 days after treatment)→87 μg/dL (21 days after treatment)) and his level of consciousness improved sufficiently for the next two days. During treatment, hepatic and renal function did not change. Although he was strongly drowsy before treatment, he began to wake up during a day and tremor also significantly improved after treatment. Twenty one days after lactulose administration, we also added intravenous branched chain amino acids (BCAA), leading to further improvement of his consciousness disorder and hyperammonemia.

## 3. Discussion

In this case, hyperammonemic encephalopathy developed in a patient with a medically intractable heart failure, who had no other apparent liver disease other than congestion. The treatment with lactulose and BCAA led to an improvement in the hyperammonemia and the encephalopathy receded following treatment for heart failure.

Consciousness disorders are sometimes complicated with heart failure [7]. Metabolic abnormalities derived from circulatory disturbance including hypoxemia, hyponatremia, or elevation of BUN, often lead to consciousness disorders in heart failure. However, these metabolic abnormalities are closely related to the circulatory disturbance that is secondary to heart failure.

On the other hand, there have been few reports on metabolic conditions where elevated serum ammonia co-occurs with heart failure. There are several case reports of hyperammonemic encephalopathy in patients with right heart failure [8,9]. Indeed, liver dysfunction is closely related to right heart failure, which could explain the association between hyperammonemic encephalopathy and right heart failure. However, the frequency of these two co-occurring is unexpectedly low, suggesting that there might be other additional factors that underlie hyperammonemic encephalopathy in heart failure patients [10,11]. Muhammad et al. suggested that infection events, in addition to right heart failure, might pose some risks for the development for hyperammonemic encephalopathy. They suggested that the occurrence of infection might aggravate the development of hyperammonemia through the effect of intestinal flora [12]. In contrast, Frea et al. presented the idea of increased ammonia by “abdominal damage” [6]. Ammonia is mainly produced in the gut as a byproduct of protein digestion [4] and bacterial metabolism [5] and it is markedly increased due to abdominal hemodynamic damage derived from low perfusion or venous congestion derived from heart failure. This condition might correspond to severe circulatory disturbance. Hence, it is important to evaluate the exact cardiac index, mean pulmonary arterial pressure, and central venous pressure by a right heart catheter in patients with hyperammonemic encephalopathy [13]. Furthermore, some medications might be a trigger of hyperammonemic encephalopathy. In the current case, diuretic drug could induce it through excessive dehydration [14].

On the other hand, ammonia might be produced by myocardial injury due to heart failure and it may be an upstream stress that worsens the heart failure, which might be attenuated by ammonia-reducing treatments. The stress, from a variety of causes (pressure overload, chemotherapy, myocardial ischemia, or others), induces myocardial injury leading to increased ammonia production from the breakdown of cellular material [15], endoplasmic reticulum stress [16], and apoptosis [17] in heart tissues in a similar way with brain tissues [18,19]. Elevated concentrations of ammonia have been shown to enhance the production of free radicals and activate inflammatory pathways through NF-κB pathway [20]. The increased tissue ammonia might decrease cell function, hence, the removal of ammonia in cases of heart failure is also clinically valuable as the treatment of heart failure.

To improve hyperammonemic encephalopathy, ammonia production needs to be reduced and any ammonia that has already been formed should be eliminated efficiently. For cases where there is no factor that can cause hepatic encephalopathy other than heart failure (as in the present case), there were hitherto no specific treatment other than the reinforcement of the heart failure therapy. However, this case suggests that lactulose (which suppresses the absorption of ammonia) and BCAA (which reduces blood ammonia concentration) may improve hyperammonemic encephalopathy. Lactulose is often used as a treatment for hepatic encephalopathy to remove ammonia and other toxins by inhibiting ammonia-producing bacteria in the intestinal tract. BCAA could also have a beneficial effect on hepatic encephalopathy by supporting ammonia detoxification in the tricarboxylic acid (TCA) -cycle in muscle tissues, leading to a reduction of plasma ammonia concentration [21,22]. In this case, lactulose and BCAA might seem to be a successful treatment strategy for hyperammonemic encephalopathy due to severe heart failure. Indeed, lactulose, which has an osmotic cathartic effect, has been reported to improve chronic renal insufficiency or heart failure by taking off excess volume [23], which might have an additional effect of reducing ammonia through the improvement of heart failure. In recent years, it has been suggested that adding albumin to lactulose could be more useful than lactulose alone for hepatic encephalopathy [24], which might also be beneficial for patients with heart failure.

Heart failure may be associated with various organ dysfunction due to hypoperfusion or congestion. This case suggests that severe right heart failure could cause the metabolic failure of ammonia, leading to hyperammonemic encephalopathy. This should be considered as a cause of impaired consciousness in severe congestive heart failure. Furthermore, this case suggests that lactulose might have dual beneficial effects in patients with hyperammonemic encephalopathy due to heart failure: improving ammonia balance and reducing circulatory volume overload (Figure 6).

In conclusion, it is important to consider the possibility of hyperammonemic encephalopathy when cases of impaired consciousness present with congestive heart failure. Serum ammonia should be measured if patients with heart failure suffer from impaired consciousness. In addition, lactulose might be effective for hyperammonemic encephalopathy secondary to heart failure.

## Figures and Tables

**Figure 1 diagnostics-10-00070-f001:**
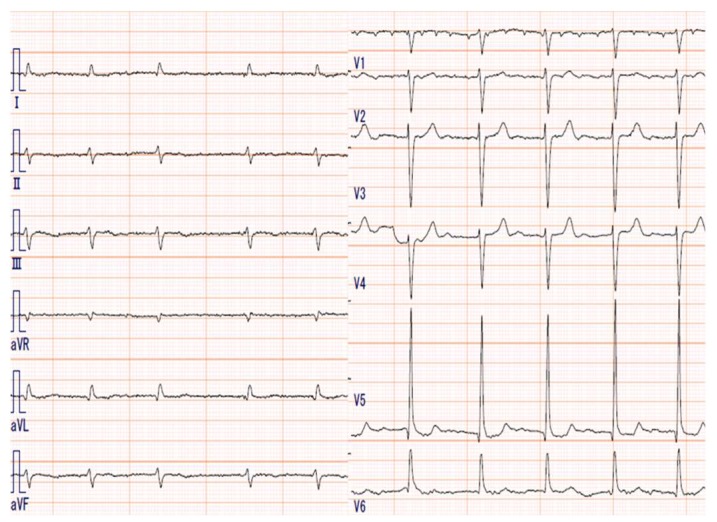
The electrocardiogram showing atrial fibrillation without any ST segmental elevation or depression.

**Figure 2 diagnostics-10-00070-f002:**
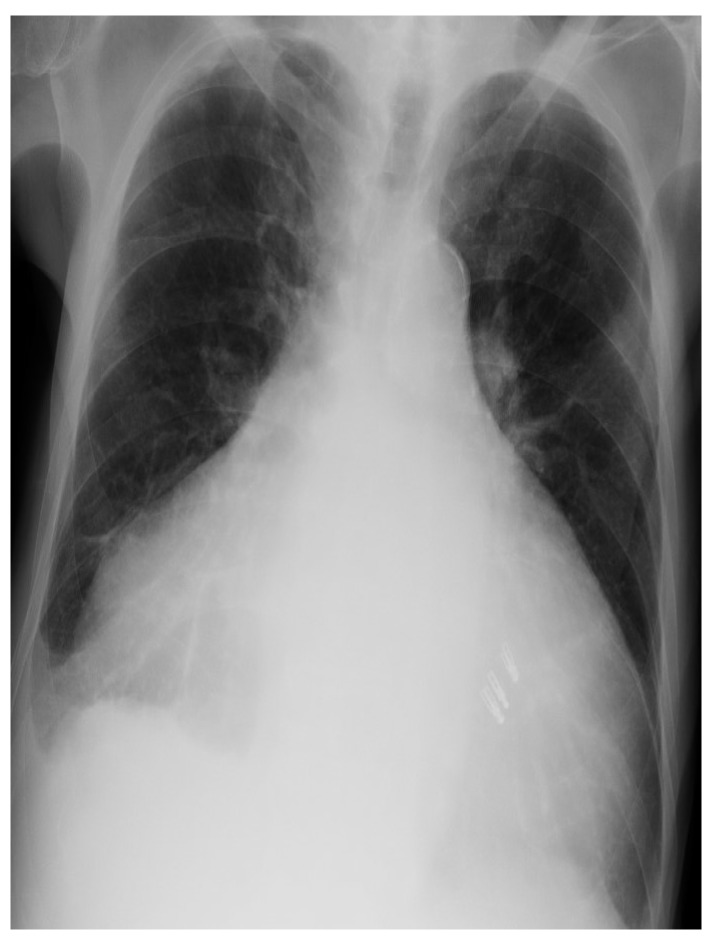
Chest x-ray showing cardiomegaly and right pleural effusion.

**Figure 3 diagnostics-10-00070-f003:**
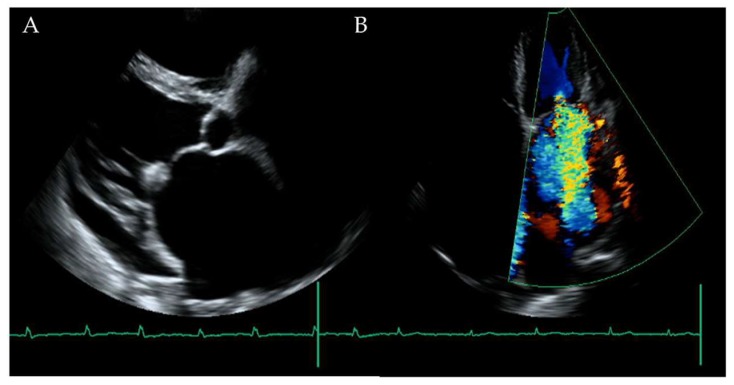
(**A**) The cardiac ultrasound showing preserved wall motion with severe left atrial dimension dilatation. (**B**) There was severe mitral regurgitation in addition to moderate tricuspid regurgitation.

**Figure 4 diagnostics-10-00070-f004:**
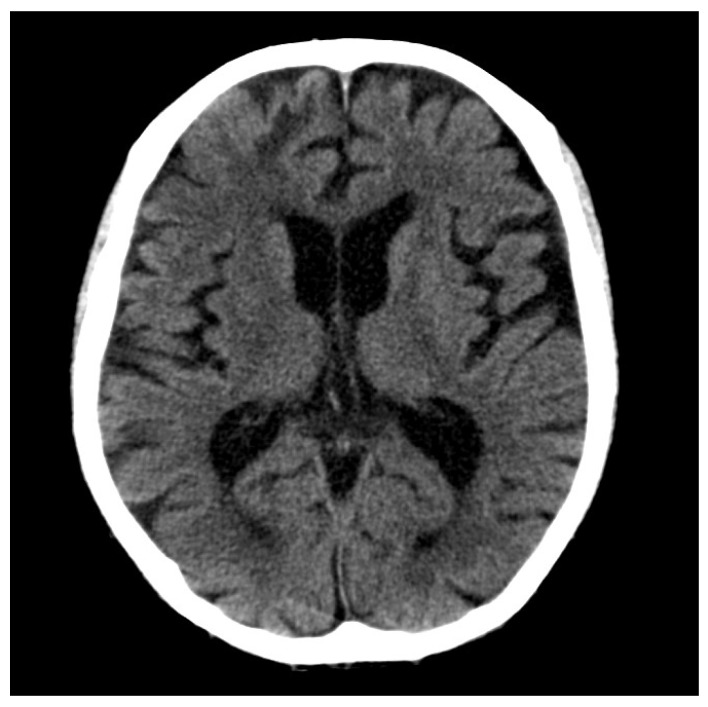
There was no significant lesion in the head computed tomography.

**Figure 5 diagnostics-10-00070-f005:**
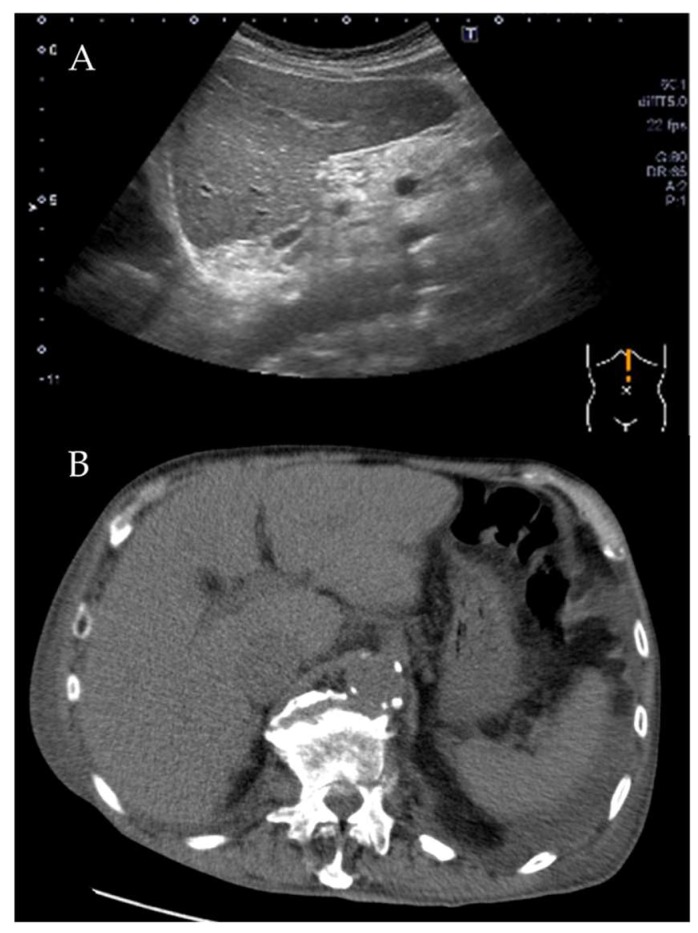
Abdominal echogram (**A**) and computed tomography (**B**) showing chronic changes due to liver congestion.

**Figure 6 diagnostics-10-00070-f006:**
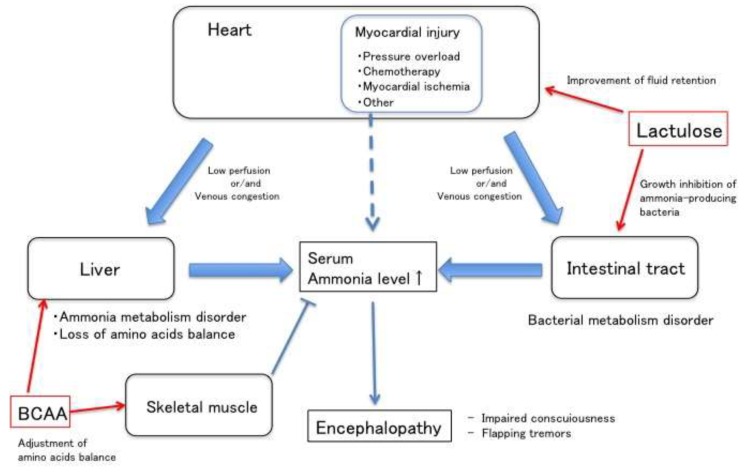
Hyperammonemia pathway in heart failure (red arrows; effective points of drugs).

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
