# Peer review of "The Efficacy of Lactulose for the Treatment of Hyperammonemic Encephalopathy Due to Severe Heart Failure"

_diagnostics, 2020, doi:10.3390/diagnostics10020070_

Round 1
Reviewer 1 Report
The efficacy of lactulose for the treatment of hyperammonemic encephalopathy due to severe heart failure, by Narita et al., submitted to Diagnostics
The following article is a case report demonstrating an 81-year old male patient with medically intractable heart failure and no apparent liver disease (cirrhosis) who presented with impaired consciousness or encephalopathy. The novelty of this case report is the fact high blood ammonia levels were reported.
The authors discuss how and why this patient developed hyperammonemia and conclude it is not due to liver disease but possibly due to increased production of ammonia in the gut from low-perfusion or venous congestion.
The case is well written and is very interesting.
I have the following comments:
It is not clear whether the lowering of blood ammonia was a result of the lactulose or the BCAA. Can you clarify. Lactulose is a non-absorbable disaccharide which acts as a laxative and is primary prescribed to treat constipation. Why was lactulose chosen to treat hyperammonemia for this patient? There is increasing evidence that ammonia is more than a neurotoxin which could act on other organs and tissues. Papers from Dr. Rose (Dasarathy et al., 2017 and Rose, 2014) discuss ammonia as a “body” metabolic stressor and therefore a few sentences should be added to the discussion mentioning these observations. Page 2; Line 70: Hepatic encephalopathy is a complication of cirrhosis and defines decompensation in liver disease patients. Since the patient did not have cirrhosis, he did not have hepatic encephalopathy. Encephalopathy was diagnosed (flapping tremors) and then the precipitating factor needed to be identified, which eventually was hyperammonemia. Therefore hyperammonemic encephalopathy is the correct term. Similar for page 5; Line 147. Please correct. Lactulose lowered blood ammonia by ~30% after 10 days and ~60% (close to ULN) after 21 days. This is quite a slow effect. Was the same dose of lactulose given throughout the 21 days? All by oral administration? Fig 6: Please describe where and how BCAA is working as stated in the discussion. Please mention the treatments which the patient was taking for heart congestion/failure. Is it possible these treatments could lead to an increase in blood ammonia? Is ammonia routinely measured in your hospital? For only patients with liver impairment? Please explain why it was decided to have ammonia measured in this patient. In the conclusion, based on the authors findings, it is worthwhile to state that blood ammonia should be measured in patients with congestive heart failure.Author Response
Response to Reviewers
We wish to express our appreciation to the reviewer. Point-by-point responses to the comments of the reviewers appear on the pages that follow. Reviewer comments are in italics, and our responses are in normal font. We wish to express our appreciation to the reviewers for his or her insightful comments, which have helped us significantly improve the paper.
Reviewer1
The following article is a case report demonstrating an 81-year old male patient with medically intractable heart failure and no apparent liver disease (cirrhosis) who presented with impaired consciousness or encephalopathy. The novelty of this case report is the fact high blood ammonia levels were reported.
The authors discuss how and why this patient developed hyperammonemia and conclude it is not due to liver disease but possibly due to increased production of ammonia in the gut from low-perfusion or venous congestion.
The case is well written and is very interesting.
I have the following comments:
It is not clear whether the lowering of blood ammonia was a result of the lactulose or the BCAA. Can you clarify.
→Thank you for your comment. Since the clinical course, ammonia had been decreasing after lactulose administration, and the consciousness level had been improved accordingly. We recognized that BCAA administered 21 days after lactulose administration contributed to further improvement of consciousness level. Therefore we thought that the main player in decreasing serum ammonia in the current case was lactulose.
We added the following words to clarify the effect derived from lactulose.
Line 84, “ Twenty one days after lactulose administration, we also added…”
Lactulose is a non-absorbable disaccharide which acts as a laxative and is primary prescribed to treat constipation. Why was lactulose chosen to treat hyperammonemia for this patient?
→Thank you for your comment. It was necessary to suppress further ammonia production in some way. Since protein restriction by decreasing protein intake could contribute to exacerbation of heart failure due to the impairment of nutritional state, we thought that suppressing ammonia production in the intestinal tract by lactulose could improve hyperammonemia without other side effects. It is converted to acetic and lactic acid, which suppress the growth and metabolism of urease-producing bacteria. Indeed, orally administered non-absorbable disaccharides such as lactulose is the first-line therapy for hyperammonemic encephalopathy.
In addition, lactulose might be favorable effect for heart failure presenting in Line 136 “ lactulose, which has an osmotic cathartic effect, …”
We added the sentences explaining the reason for the use of lactulose as follows
Line 79, “ in order to suppress ammonia production in the intestinal tract”.
There is increasing evidence that ammonia is more than a neurotoxin which could act on other organs and tissues. Papers from Dr. Rose (Dasarathy et al., 2017 and Rose, 2014) discuss ammonia as a “body” metabolic stressor and therefore a few sentences should be added to the discussion mentioning these observations.
→Thank you for your valuable comments pointing out the discussion. We added the comment about it in discussion as follows and we cited the manuscript by Dr Dasarathy and Rose.
Line133
“Elevated concentrations of ammonia have been shown to enhance the production of free radicals and activates inflammatory pathways through NF-kB pathway.”
Page 2; Line 70: Hepatic encephalopathy is a complication of cirrhosis and defines decompensation in liver disease patients. Since the patient did not have cirrhosis, he did not have hepatic encephalopathy. Encephalopathy was diagnosed (flapping tremors) and then the precipitating factor needed to be identified, which eventually was hyperammonemia. Therefore hyperammonemic encephalopathy is the correct term. Similar for page 5; Line 147. Please correct.
→Thank you for your comments. We have changed the word, “hepatic encephalopathy with hyperammonemia”.
Lactulose lowered blood ammonia by ~30% after 10 days and ~60% (close to ULN) after 21 days. This is quite a slow effect. Was the same dose of lactulose given throughout the 21 days? All by oral administration?
→Thank you for your question. The answer is “Yes”. He had been receiving the same dose for 21 days by oral administration because there had been no evidence for the use of lactulose against hyperammonemia due to advanced heart failure We do not know about the speed at which the level of serum ammonia is decreased by oral administration of lactulose in the setting of heart failure. However, if we checked the level of serum ammonia more frequently, more exact speed of the improvement in the level of serum ammonia could be examined.
Fig 6: Please describe where and how BCAA is working as stated in the discussion.
→Thank you for your comments. BCAA could have a beneficial effect on hepatic encephalopathy by supporting ammonia detoxification in the TCA-cycle in muscle tissues, leading to a reduction of plasma ammonia concentration (Selorm S et al J Gastro Hepato 34 (2019)31-39). In addition, it stimulates hepatic protein synthesis, inhibit protein degradation (Freund H, et al Ann. Surg 190 1979 18-23) We corrected Fig 6 and added the skeletal muscle pathway which is affected by BCAA in revised version.
Please mention the treatments which the patient was taking for heart congestion/failure. Is it possible these treatments could lead to an increase in blood ammonia?
→Thank you for your comments. He was taking beta-blockers and diuretics to treat congestive heart failure. There were some reports that diuretic therapy might enhance the risk of hyperammonemia induced by hypovolemia (17. Häussinger D, Kaiser S, Stehle T, Gerok W. Liver carbonic anhydrase and urea synthesis. The effect of diuretics. Biochem Pharmacol. 1986, 35, 3317-22.). We added the comment about it in Line 126.
Is ammonia routinely measured in your hospital? For only patients with liver impairment? Please explain why it was decided to have ammonia measured in this patient.
→Thank you for your valuable comments. We do not routinely measure ammonia of patients with heart failure in our hospital. In this case, since tremors and unconsciousness was observed, ammonia had been measured in consideration of hyperammonemic encephalopathy as a differential diagnosis.
In the conclusion, based on the authors findings, it is worthwhile to state that blood ammonia should be measured in patients with congestive heart failure.
→Thank you for your precious comments. We believe that it is important to consider the possibility of hyperammonemic encephalopathy in case of impaired consciousness present with advanced heart failure.
We added the comment about it in conclusion.
“Serum ammonia should be measured if patients with heart failure suffer from impaired consciousness.”
Reviewer 2 Report
The paper entitled “The Efficacy of Lactulose for the Treatment of Hyperammonemic Encephalopathy due to Severe Heart Failure”, reported on an interesting case report on the therapeutic use of Lactulose for the Treatment of Hyperammonemic Encephalopathy.
The findings are well described, given further information regarding some therapeutical approach in case of presence of neurological signs related to hyperammonemia. However the manuscript presented few methodological concerns that should be clarified; a more detailed laboratory assessments description and a more detailed cardiologic finding could be useful reported to confirm the relation between the hyperammonemic encephalopathy and the heart failure.
Specific concerns:
Case Presentation
Line 45 “A years ago”. Please add the word “one” instead of “a”.
Please add further information about the characteristics of laboratory assessments related to a probably hepatic viral infection: serum HCV-RNA and serum HBV RNA values, C-Reactive Protein (CRP), and other signs of infection (blood counts; Platelet counts); a detailed cerebral MRI if performed, could be added;
Line 63-64: “Echocardiogram demonstrated preserved ejection fraction and severe MR in addition to moderate tricuspid regurgitation”.
The authors suspected the role of a probably right heart failure in the onset of hepatic congestion, but no exam confirmed this relation but for the moderate tricuspid regurgitation. Please it could be important to report TAPSE (tricuspid anular plane systolic excursion) and the TDI values to better understand the function of right ventricle involvement.
Author Response
Response to Reviewers
We wish to express our appreciation to the reviewer. Point-by-point responses to the comments of the reviewers appear on the pages that follow. Reviewer comments are in italics, and our responses are in normal font. We wish to express our appreciation to the reviewers for his or her insightful comments, which have helped us significantly improve the paper.
Reviewer2
The paper entitled “The Efficacy of Lactulose for the Treatment of Hyperammonemic Encephalopathy due to Severe Heart Failure”, reported on an interesting case report on the therapeutic use of Lactulose for the Treatment of Hyperammonemic Encephalopathy.
The findings are well described, given further information regarding some therapeutical approach in case of presence of neurological signs related to hyperammonemia. However the manuscript presented few methodological concerns that should be clarified;
a more detailed laboratory assessments description and a more detailed cardiologic finding could be useful reported to confirm the relation between the hyperammonemic encephalopathy and the heart failure.
Thank you for your precious comments. As reviewer#2 commented, more concise description of this case might support the relationship between hyperammonemic encephalopathy and heart failure. Therefore we added the concise laboratory data and hemodynamic data of right heart catheterization performed six months before his admission in Line 48.
However, it is difficult to relate his hyperammonemic encephalopathy with heart failure, because we conclude indirectly that his encephalopathy was due to heart failure because there was no evidence of primary liver disease other than liver congestion. Further studies investigating about the association between hyperammonemic encephalopathy and heart failure would be warranted.
Line 45 “A years ago”. Please add the word “one” instead of “a”.
→Thank you for your comments. We corrected the word in revised manuscript.
Please add further information about the characteristics of laboratory assessments related to a probably hepatic viral infection: serum HCV-RNA and serum HBV RNA values, C-Reactive Protein (CRP), and other signs of infection (blood counts; Platelet counts); a detailed cerebral MRI if performed, could be added;
→Thank you for your comments. We added the data of hepatitis B and C (HBV and HCV) in Line 75. However, I’m sorry that we did not measure serum HCV-RNA and serum HBV RNA values. We presented other laboratory data such as C reactive protein 0.21mg/dL, white blood cell counts 3000/μL, Platelet counts 82000/μL. We did not perform cerebral MRI by technical reason. We added these data in revised manuscript.
Line 63-64: “Echocardiogram demonstrated preserved ejection fraction and severe MR in addition to moderate tricuspid regurgitation”.
→Thank you for your comments. For heart enlargement, bi-atrial chamber expanded by atrial fibrillation, and functional MR and TR had been observed.
The authors suspected the role of a probably right heart failure in the onset of hepatic congestion, but no exam confirmed this relation but for the moderate tricuspid regurgitation. Please it could be important to report TAPSE (tricuspid anular plane systolic excursion) and the TDI values to better understand the function of right ventricle involvement.
→Thank you for your comments. We performed echocardiogram, TAPSE 18mm and the TDI values RV S’ 12.4 cm/s, RV E’ 9.0cm/s, septal E/E’ 21.7. We consider that chronic congestive liver damage is probably due to right heart failure. We added these data in revised manuscript Line 68.
Round 2
Reviewer 2 Report
No comments to do